# Improvement of Blood Flow and Epidermal Temperature in Cold Feet Using Far-Infrared Rays Emitted from Loess Balls Manufactured by Low-Temperature Wet Drying Method: A Randomized Trial

**DOI:** 10.3390/biomedicines13071759

**Published:** 2025-07-18

**Authors:** Yong Il Shin, Min Seok Kim, Yeong Ae Yang, Yun Jeong Lee, Gye Rok Jeon, Jae Ho Kim, Yeon Jin Choi, Woo Cheol Choi, Jae Hyung Kim

**Affiliations:** 1Department of Rehabilitation Medicine, School of Medicine, Pusan National University, Yangsan 50612, Republic of Korea; rmshin@pusan.ac.kr; 2Department of Rehabilitation Medicine, Yangsan Hospital, Pusan National University, Yangsan 50612, Republic of Korea; 3Monash Health, Melbourne, VIC 3800, Australia; minseok.kim@monashhealth.org; 4Department of Occupational Therapy, Inje University, Gimhae 50834, Republic of Korea; otyya62@inje.ac.kr (Y.A.Y.); wmfghi69@naver.com (Y.J.L.); 5R&D Center, eXsolit, Yangsan 50611, Republic of Korea; grjeon@pusan.ac.kr (G.R.J.); jhkim@pusan.ac.kr (J.H.K.); 6R&D Center, Hanwool Bio, Yangsan 50516, Republic of Korea; hbio1004@naver.com (Y.J.C.); lih1769@naver.com (W.C.C.)

**Keywords:** cold feet, loess bio-balls, far-infrared rays (FIR), body’s core temperature, thermoregulation

## Abstract

**Background:** Cold feet syndrome is characterized by hypersensitivity of sympathetic nerves to cold stimuli, resulting in vasoconstriction and reduced peripheral blood flow. This condition causes an intense cold sensation, particularly in the extremities. Although hormonal changes (e.g., during childbirth or menopause) and psychological stress have been implicated, the mechanisms and effective treatments remain unclear. **Methods:** Ninety adult volunteers were randomized into three groups based on the type of heating mat applied to the feet, with surface temperatures gradually increased from 20 °C to 50 °C. Group A (control) used non-FIR electric mats, Group B used carbon FIR mats, and Group C used loess bio-ball FIR mats. Blood flow (mL/min/100 g) and epidermal temperature (°C) in the left big toe (LBT) and right big toe (RBT) were measured before and after heating or FIR exposure using laser Doppler flowmetry and infrared thermometers. **Results:** No significant changes in blood flow or skin temperature were observed in Group A. In Group B, blood flow increased by 15.07 mL/min/100 g in the LBT (from 4.12 ± 2.22 to 19.19 ± 5.44) and by 14.55 mL/min/100 g in the RBT (from 4.26 ± 2.29 to 18.81 ± 4.29). In Group C, blood flow increased by 32.86 mL/min/100 g in the LBT (from 4.23 ± 1.64 to 37.09 ± 6.04) and by 32.63 mL/min/100 g in the RBT (from 4.20 ± 1.61 to 36.83 ± 6.48). Epidermal temperature also increased significantly in Group C. All changes in Groups B and C were statistically significant (*p* < 0.05), with Group C showing the most prominent enhancement. **Conclusions:** The loess bio-ball mat significantly increased both peripheral blood flow and epidermal temperature compared to the electric and carbon mats. These findings suggest that FIR emitted from loess bio-balls may enhance peripheral circulation through hypothalamus thermogenic response and nitric oxide (NO)-dependent pathways and could serve as a complementary and non-invasive intervention for individuals with poor blood flow.

## 1. Introduction

Cold feet are believed to result from hypersensitivity of the sympathetic nervous system to cold stimuli, which induces vasoconstriction and reduces blood flow to the peripheral regions, particularly the hands and feet [1,2]. This reduction in peripheral circulation leads to a pronounced sensation of cold. Moreover, cold feet may be associated with various pathological conditions including impaired immune function, gastrointestinal disorders, arteriosclerosis, anemia, hypertension, and autonomic nervous system dysfunction. The management of cold extremities typically focuses on enhancing peripheral circulation and maintaining core body temperature. Interventions such as avoiding cold exposure, consuming warm foods and beverages, and using foot baths or half-baths have been shown to improve blood flow. Regular aerobic exercise and stress reduction are also known to be beneficial. However, despite these strategies, the underlying etiology of cold feet—particularly in Raynaud’s syndrome, a condition closely related to cold extremities—remains unclear [3,4].

Far-infrared (FIR) therapy utilizes electromagnetic radiation that penetrates the epidermis to increase the body temperature, improve blood circulation, and activate various metabolic processes [5]. FIR with wavelengths between 4 to 16 μm can penetrate up to 4–5 cm into the skin, where it stimulates the vibrational motion of water molecules and generates heat [6]. FIR is believed to enhance fluid mobility by affecting water clusters in vivo [7], and its biological effects may be amplified by resonance, as its frequency range overlaps with the natural resonance frequency of water molecules [8]. FIR therapy has demonstrated various therapeutic effects and has been applied to the treatment of a wide range of diseases and symptoms [9,10]. It has been reported to reduce muscle damage and accelerate recovery after physical activity [11]. In patients with peripheral vascular disease, FIR has been shown to improve skin temperature, peripheral blood flow, and plantar pressure [12]. Furthermore, FIR therapy has been found to increase epidermal temperature and enhance sensory perception, including sensitivity to pain, touch, and pressure, particularly in patients with diabetic foot. FIR lamp therapy has also been investigated for its potential to reduce muscle damage and aid recovery in athletes [13]. Moreover, FIR has been associated with beneficial effects on blood pressure regulation and lipid metabolism, which may contribute to a reduced risk of cardiovascular diseases [14]. Low-energy FIR is widely used in the management of vascular-related conditions; however, its precise mechanism of action remains insufficiently understood [14,15].

Compared to raw loess, loess powder that has been heat-treated at high temperatures (850–1050 °C) exhibits significantly reduced FIR absorption in the 9.5–9.8 μm range [16]. This indicates that raw loess effectively emits FIR within this wavelength range, which corresponds to the absorption spectrum of water molecules. As a result, FIR emitted from raw loess may be absorbed by the body’s water content, potentially producing biological effects. Accordingly, this study was designed as a comparative experiment to identify the most effective method for promoting blood circulation and warming the feet in individuals with cold feet. Specifically, a comparison was made between a conventional conductive heating mat and a new technology (a loess bio-ball mat) to see which method was more effective in increasing blood flow and skin temperature in the toes.

## 2. Materials and Methods

### 2.1. Infrared Absorption Spectra

According to the infrared (IR) absorption spectra of loess powders prepared under various conditions, loess powders exhibit several notable properties [17]. Two characteristic peaks were observed around 3600–3700 $\;{cm}^{-1}$ (2.6–2.7  μm) and 1005–1105 ${cm}^{-1}$ (9.5–9.8 μm) in untreated loess powder. When the loess powder was heated at 850 °C for 2 h or at 1050 °C for 1 h, the peaks disappeared. Additionally, a large absorption peak appeared around 9.4–9.8 μm in the heat-treated loess powder. This is due to the vibrational motion (stretching) of Si-O bonds and indicates that the raw loess emits a significant amount of FIR at this wavelength. The IR absorption spectra demonstrated that the unheated loess powder retained the functional properties of the beneficial minerals originally present in the raw loess. However, these properties degrade significantly during high-temperature processing. In conventional methods, loess balls are manufactured by baking loess powder at a high temperature (1000–1200 °C). In this study, a new type of loess bio-ball was developed using a low-temperature wet drying method, maintaining the temperature below 90 °C [16]. Consequently, the functional loess ball produced using this new method is referred to as a loess ball manufactured by low-temperature wet drying method or a loess bio-ball.

### 2.2. Manufacturing of Loess Balls Using Low-Temperature Wet Drying Method

X-ray diffraction (XRD) analysis of the loess powder confirmed the presence of peaks corresponding to kaolinite (Al_2_SiO_2_O_5_(OH)_4_) and quartz (SiO_2_) [16]. Table 1 summarizes the mineral composition of the loess powder. The primary mineral components included feldspar and quartz, while the clay materials were mica and kaolin.

The loess bio-balls were manufactured using a method that utilizes the beneficial properties of minerals and microbial enzymes inherent in raw loess. Unlike the conventional approach of heat-treating loess at high temperatures, the loess bio-balls in this study were produced by the wet drying method at temperatures below 90 °C. The detailed manufacturing process has been described in the previously published literature [16]. The raw loess material was excavated from the Sancheong area in Gyeongnam Province, Korea. The original loess powder was aged at low temperatures for more than 6 months. The matured loess was then formed into balls using a ball machine (SMP-360, Jeil Pharmaceutical Machinery Manufacturing Co., Ltd., Daegu, Korea) and a pill-making machine (SMP-660, Jeil Pharmaceutical Machinery Manufacturing Co., Ltd., Daegu, Korea) without heating or pressurization. The IR absorption spectra of raw loess and high-temperature heat-treated loess powder, the emission spectra of loess balls, and the temperature-dependent radiant energy and transmittance characteristics of loess bio-balls to various materials have been described in detail in previously published studies [16].

### 2.3. Energy Conversion Between FIR Emitted from Loess Bio-Balls and Water Molecules

The functional loess bio-balls exhibited radiant intensity in the wavelength of 5–20 μm at 40 °C (313 K). The peak radiant intensity was measured at $3.74\times 10^{2}\;W/m^{2}$, centered around 9.5–9.8 μm. This FIR was attributed to Si-O stretching vibrations within the loess bio-balls. At room temperature, the oxygen (O) and hydrogen (H) atoms in water molecules undergo three fundamental vibrational modes: symmetric stretching ($v_{1}=3280\;{cm}^{-1})$, bending ($v_{2}=1564\;{cm}^{-1})$, and asymmetric stretching ($v_{2}=3490\;{cm}^{-1})$ [18]. These vibrational motions enable water molecules to efficiently absorb FIR in the 2.7–20 μm wavelength range. Notably, the emission wavelength range (5–20 μm), particularly around 5.6–14 μm—referred to as the “growth line”—closely matches the absorption range of water, which is considered critical for supporting the physiological activity of living organisms [19,20]. It is well established that FIR wavelengths around 9.4 μm stimulate intracellular heat shock proteins (HSPs) and nitric oxides (NO) production pathways, leading to increased endothelial nitric oxide synthase (eNOS) expression and relaxation of vascular smooth muscles. This results in vasodilation and improved blood flow in capillaries and peripheral blood vessels. Therefore, the FIR emitted from loess bio-balls can be selectively absorbed by water molecules in biological tissues, primarily as thermal energy via wave resonance. The remaining energy is transferred as vibrational energy, potentially activating surrounding biological tissues [8,21].

### 2.4. Systemic and Local Thermoregulatory Mechanisms in the Human Body

#### 2.4.1. Mechanisms of Thermoregulation in the Body

The hypothalamus in the brain functions as the regulatory center for body temperature control and acts as the body’s automatic thermostat. Thermal changes are detected by receptors located within the hypothalamus [22,23]. Through the thermoregulatory center of the hypothalamus, a stable core body temperature of approximately 37 °C (98.6 F) is maintained, which is optimal for enzymatic activity [24,25]. In addition, the skin contains thermoreceptors that provide feedback signals to the hypothalamic center, thereby assisting in thermoregulation [23]. When the body’s core temperature drops below 37 °C, receptors signal the hypothalamus, which then transmits impulses to the arterioles in the skin. This induces vasoconstriction, reducing capillary blood flow and minimizing heat loss to the environment, as illustrated in Figure 1a [26,27]. Conversely, when the core temperature rises above 37 °C, receptors again signal to the hypothalamus, which promotes vasodilation by sending impulses to the skin arterioles. This results in increased capillary blood flow, which enhances heat dissipation, as shown in Figure 1b, When the external temperature is high and the body’s heat loss mechanism is insufficient, the hypothalamus also stimulates sweat gland activity to promote evaporation cooling, thereby reducing core body temperature.

Cold feet is a condition characterized by abnormally cold and dull sensations in the feet, even in environments that are not objectively cold. This occurs when the sympathetic nerves become hypersensitive to cold stimuli, leading to vasoconstriction and a subsequent reduction in blood supply to the feet. Various factors may contribute to this condition, including underlying medical conditions and lifestyle habits. A therapeutic approach for cold feet involves whole-body irradiation using FIR, the wavelength of which closely matches the resonance wavelength of water molecules. FIR exposure increases the core body temperature above 37 °C. In response, the hypothalamus activates thermoregulatory mechanisms, promoting vasodilation and increasing blood flow to the peripheral capillaries, particularly in the feet, as illustrated in Figure 1b. Consequently, toe temperature rises due to improved peripheral circulation induced by FIR therapy.

#### 2.4.2. TRPV1 Receptor-Mediated Axon Reflexes

Transient receptor potential vanilloid 1 (TRPV1) is a calcium-permeable ion channel primary activated by noxious heat and capsaicin, the pungent compound in red chili peppe, as well as other vanilloid-related chemicals [28]. It plays an key neurophysiological role in mediating rapid, local inflammatory responses to harmful stimuli. Upon activation by noxious heat or pain, TRPV1 transmits signals retrogradely not only to the spinal cord but also to nearby nerve endings, inducing the release of neuropeptides. This triggers local vasodilation and edema (neurogenic inflammation). TRPV1 channels open in response to stimuli such as temperature above 43 °C, capsaicin, or acidic substances with low pH, allowing Na^+^ and Ca^2+^ to enter the nerve cells and generate action potentials (electrical signals). Unlike general reflexes involving central pathways, axonal reflexes are localized and occur within the myenteric nervous system without involvement the brain and spinal cord.

The step-by-step process of the TRPV1-mediated axonal reflex is as follows. Step 1 (stimulus and TRPV1 activation): When the skin is exposed to a hot object, inflammatory mediators from a wound, or capsaicin, TRPV1 receptors in the sensory nerve endings of the affected area are activated. Step 2 (action potential generation and bidirectional propagation): When TRPV1 channels open, nerve cells are excited and generate an action potential. In the forward (orthodromic) direction, the signal is transmitted along the axon to the spinal cord and then to the brain, where it is perceived as pain or heat. In the reverse (antidromic) direction, the signal travels backward through the axon branches to other peripheral nerve endings. Step 3 (neuropeptide secretion): Neuropeptides such as CGRP (calcitonin gene-related peptide) and substance P are secreted at peripheral nerve endings where the retrograde signals arrive. Step 4 (local neurogenic inflammatory response): The released neuropeptides act on the surrounding tissues to induce an inflammatory response. When CGRP is released, it causes relaxation of the surrounding capillary smooth muscles, resulting in vasodilation and increased blood flow, which makes the skin appear red and warm. When substance P is released, it increases vascular permeability by widening the gaps between endothelial cells in the blood vessel wall, allowing plasma components to leak into surrounding tissues and causing local swelling (wheal).

#### 2.4.3. NO-Dependent Vasodilation

Nitric oxide (NO), a signaling molecule, plays an important physiological role in regulating blood pressure and blood flow by promoting the relaxation and dilation of blood vessels [29]. The biochemistry of NO is complex, and new insights into the regulation of NO biosynthesis and signaling mechanisms are continuously emerging. NO serves as a key regulator of cardiovascular function, metabolism, neurotransmission, immunity, and more. Abnormal NO signaling is a hallmark of several major diseases, including cardiovascular disease, diabetes, and cancer [30]. The mechanism of NO action proceeds as follows. Step 1 (stimulation): Physical stimuli, such as shear stress from blood flow, generate frictional forces on the vascular endothelium. Chemical stimuli include the binding of neurotransmitters such as acetylcholine and bradykinin or hormones to specific receptors on the vascular endothelium. Step 2 (NO production): Upon stimulation, the intracellular calcium (Ca^2+^) levels increase within vascular endothelial cells. Elevated intracellular calcium activates endothelial nitric oxide synthase (eNOS), which catalyzes the conversion of L-arginine into NO and L-citrulline. Step 3 (NO diffusion): The generated NO is a small gaseous molecule that easily diffuses into the adjacent vascular smooth muscle cells. Step 4 (signal transduction in smooth muscle): In smooth muscle cells, NO activates the enzyme soluble guanylyl cyclase (sGC). Activated sGC is converted into cyclic guanosine monophosphate (cGMP), a secondary signaling molecule. Elevated levels of cGMP initiate a signaling cascade in smooth muscle cells, promoting relaxation and vasodilation. Stage 5 (relaxation and vasodilation): Increased cGMP levels reduce intracellular calcium concentration via several pathways, including the activation of protein kinase G (PKG). As a result, smooth muscle contraction is reduced. This relaxation allows the blood vessel walls to expand, resulting in vasodilation.

### 2.5. Participants

Participant recruitment for clinical research on blood circulation was announced in an article published in the local newspaper Yangsan News Park on 7 November 2023. Additional participants were recruited following the publication of an announcement in the Yangsan News Park on 26 December 2024, for subsequent trials, primarily involving comparative studies using carbon mats. Eligibility criteria included individuals 30 years or older who frequently experienced discomfort due to cold extremities or symptoms of blood circulation disorders. Volunteers with severe heart disease, cancer, or mental illness (including dementia), or who were pregnant, were excluded. Individuals taking medications that could affect autonomic nervous system (ANS) function were also excluded. After a thorough explanation of the study protocol, informed consent was obtained from all participants. This study was conducted in accordance with the principles of the Declaration of Helsinki.

Participant demographics and baseline characteristics of the 90 individuals who ultimately participated and were included in the analysis are summarized in Table 2. For control Group A (16 females and 14 males), the mean age was 57.44 ± 8.53 years, mean height was 165.12 ± 8.65 cm, mean body weight was 63.70 ± 9.19 kg, and mean BMI was 23.36 ± 3.26 $kg/m^{2}$. For the experimental Group B (15 females and 15 males), the mean age was 59.24 ± 6.82 years, mean height was 164.90 ± 9.38 cm, mean body weight was 65.41 ± 9.03 kg, and mean BMI was 24.05 ± 2.40 $kg/m^{2}$. For the experimental Group C (14 females and 16 males), the mean age was 60.17 ± 10.35 years, mean height was 163.70 ± 9.19 cm, mean body weight was 63.66 ± 6.79 kg, and mean BMI was 23.76 ± 2.54 $kg/m^{2}$. There were no statistically significant differences in age, height, body weight, or BMI among the three groups (*p* > 0.05). However, Group A showed slightly higher baseline skin temperature and blood flow compared to both experimental groups, with small to medium effect sizes (Cohen’s d = 0.25–0.48).

For three-group comparisons, ANOVA with η^2^ or partial η^2^ was used. Data are presented as the mean ± standard deviation (SD). Body mass index (BMI) was calculated as body weight (kg) divided by height squared (m^2^). LBT stands for the left big toe, and RBT stands for the right big toe.

### 2.6. Trial Design and Setting

The study protocol was approved by the Inje University Bioethics Committee (approval number: INJE 2023-05-035-005) on 20 September 2023, as a clinical trial titled “Improvement of blood circulation and health promotion effects of related diseases by far-infrared rays emitted from loess bio-balls.” The Inje University Bioethics Committee also approved an additional protocol for further experiments (approval number: INJE 2023-05-035-007) on 31 December 2024. The study was conducted primarily at Hanwool Bio Lab, Yangsan-si, South Korea, and followed the Consolidated Standard of Reporting Trials (CONSORT) guidelines [31]. In the intervention experiment, bio-signal measurements were performed by two researchers with occupational therapy qualifications from the Department of Occupational Therapy at Inje University (both with doctoral degrees, one with more than 5 years of experience and the other with more than 10 years of experience).

As shown in Figure 2, 105 participants initially volunteered to participate in the blood flow-related experiment. Of these, five were excluded because they did not meet the eligibility criteria, and two withdrew for personal reasons. The study administrator was in charge of the random assignment sequence. Participants were randomly assigned to one of the following three groups using a sealed-envelope randomization method that included application forms for experiments of types A, B, and C. Group A used an electric mat, Group B used a carbon mat that emitted FIR in a linear shape from carbon coated with heat wires, and Group C used a loess bio-ball mat that emitted FIR in an area shape from loess bio-balls. Cushions and covers were used on top of the three mats so that the participants did not know the type of mat they were using. Furthermore, the measures were also unaware of the participants’ health histories and conditions (double blinding). After randomization, 98 participants were included in the study. Of these, 33 were assigned to Group A, 32 to Group B, and 33 to Group C. During follow-up, one participant in Group A dropped out due to illness and two withdrew for personal reasons, resulting in 30 participants. In Group B, two participants dropped out for personal reasons, resulting in 30 participants. In Group C, two participants dropped out due to illness and one withdrew for personal reasons, resulting in 30 participants. Thus, 90 participants completed the study, and their experimental results were included in the final analysis. The sample size was set at a minimum of 30 per group to ensure adequate statistical power at a 0.05 level (*p* < 0.05). The personnel responsible for enrolling and assigning participants were blinded to the randomization sequence. Since this was personal information, no one—including participants, treatment providers, evaluators, and data analysts—was identified by their real names after the group assignment.

### 2.7. Interventions

A conductive electric mat (SW-301, SHS; Samhwa Electric Blanket, Pusan, Korea) was used in control Group A. The temperature was set at 3 °C intervals, ranging from 20 °C to 50 °C. For comparative experiments, carbon graphene mats (DEB-5203, Daejin Electronics Co., Ltd., Yongin, Korea), which emit FIR, were used in experimental Group B. For experimental Group C, a loess bio-ball bed (7111, Jangsoo Co. Ltd., Seoul, Korea) with a radiant intensity of $3.74\times 10^{2}\;W/m^{2}$ and an emission wavelength of approximately 9.5–9.8 μm, was used as the FIR source, with temperature increments (1 °C interval) from 20 °C to 50 °C. Blood flow was measured using a Doppler laser flow meter (ALF21; Advanced Co., Ltd., Tokyo, Japan). When measuring blood flow using the Advance Laser Flowmeter AFL21, the product specifications are as follows: number of optical probes: 1; measurement mode: continuous perfusion; duration of individual measurements: 30 s per reading; and applied signal post-processing procedure: average of 5 measurements and subtraction of zero offset. A red diode laser (780 nm, 2 mW) applied to the reflective electrode can penetrate approximately 0.5–1 mm beneath the skin. The electrode was attached to the LBT and RBT using a bandage. Blood flow and skin temperature were measured in both big toes while increasing the set temperature at 3 °C intervals (or 1 °C intervals in Group C) from 20 to 50 °C. Epidermal temperature was measured using an infrared thermometer (MM-GP100, Harbin Xiande Technology Development Co., Ltd., Shenzhen, China) and a non-contact industrial infrared digital thermometer (LUAZ-GM320, Delona partner, Shenzhen, China).

Data analysts (evaluators) conducted the assessment without knowing the participants’ health statuses or group assignments (blinded evaluation). Figure 3a shows a photograph of a reflective electrode attached to the right big toe for blood flow measurement on a loess bio-ball mat, and Figure 3b shows a photograph of the blood flow measurement on the left big toe using a cushion placed on the mat to minimize conductive emission.

The loess bio-ball mat contains an internal insulation structure designed to minimize heat conduction from the internal heating wire [17]. In contrast, the electric mat and carbon mat were expected to potentially cause thermal discomfort or even thermal damage to participants due to higher heat conduction. To ensure consistent thermal conditions across all mats, a cushion approximately 1.5–2 cm thick was placed on top of each mat. Subsequently, each participant’s blood flow and skin temperature were measured under similar standardized conditions. To ensure participant safety, the surface temperature of the cushion was continuously monitored throughout the measurement and maintained below 38 °C.

### 2.8. Statistical Analyses

All statistical analyses for control Group A (*n* = 30), experimental Group B (*n* = 30), and experimental Group C (*n* = 30) were conducted using IBM SPSS Statistics (version 29.0.2.0; IBM Corp., SPSS Inc., Chicago, IL, USA). The Shapiro–Wilk test was used to assess the normality of the data distribution. Since the test results indicated that the data were not normally distributed, Spearman’s rank correlation coefficient was used instead of the Pearson correlation coefficient. Spearman correlation coefficients (r) and *p*-values were calculated for blood flow and epidermal temperature based on the set temperatures of the mats. Data processing, graphing, and logistic curve fitting were performed using Microsoft Excel 2016 (Microsoft Corp., Redmond, WA, USA).

## 3. Results

### 3.1. Changes in Blood Flow and Epidermal Temperature Measured in LBT and RBT When Using an Electric Mat

Table 3 presents the measurements of blood flow and epidermal temperature in the LBT and RBT using a non-far-infrared (non-FIR) electric mat (SW-301, SHS; Samhwa Electric Blanket, Pusan, Korea). When the mat’s set temperature was increased from 20 °C to 50 °C, there was minimal change in blood flow in the LBT (0.10 mL/min/100 g, 2.10%). The epidermal temperature in the LBT showed a slight increase of 0.54 °C (1.84%). Similarly, blood flow in the RBT showed only a slight change (0.16 mL/min/100 g, 3.39%), and epidermal temperature in the RBT increased slightly, by 0.48 °C (1.64%). These results suggest that the electric mat used as a placebo mat has minimal effects on blood flow and skin temperature in the context of cold feet.

Spearman’s correlation coefficients (r) for blood flow (BF) at set temperature intervals on the electric mat ranged from 0.999 (20–23 °C) to 0.937 (20–35 °C) in the LBT (*p* < 0.001) and from 0.987 (23–26 °C) to 0.881 (29–47 °C) in the RBT (*p* < 0.001). The r values for skin temperature (ST) at set temperature intervals on the electric mat ranged from 0.978 (23–26 °C) to 0.881 (29–47 °C) in the LBT (*p* < 0.001) and from 0.995 (20–23 °C) to 0.838 (32–50 °C) in the RBT (*p* < 0.001). The very high correlation coefficient (>0.9) suggests that blood flow and skin temperature tend to increase together with temperature changes. However, because the actual changes in blood flow and skin temperature were small, even if the correlation was statistically high, the clinically significant effect was small.

### 3.2. Changes in Blood Flow and Epidermal Temperature Measured in LBT and RBT When Using the Carbon Mat (Line-Emitting FIR)

Table 4 presents the measurements of blood flow and epidermal temperature in the LBT and RBT using a carbon graphene mat emitting line-type FIR. When the set temperature applied to the mat was increased from 20 °C to 50 °C, blood flow in the LBT increased by 15.05 mL/min/100 g, from 4.12 ± 2.22 mL/min/100 g to 19.17 ± 5.44 mL/min/100 g. The epidermal temperature in the LBT gradually increased by 5.95 °C (20.80%), from 28.61 ± 1.50 °C to 34.56 ± 1.26 °C. Similarly, blood flow in the RBT increased by 14.55 mL/min/100 g, from 4.26 ± 2.29 mL/min/100 g to 18.81 ± 4.77 mL/min/100 g. The epidermal temperature in the RBT increased by 6.04 °C (21.18%), from 28.52 ± 1.24 °C to 34.56±1.34 °C. These results indicate that a carbon mat emitting FIR enhances both blood flow and skin temperature, which may help alleviate symptoms of cold extremities, particularly in the feet.

Spearman’s correlation coefficients (r) for BF at set temperature intervals on the carbon mat ranged from 0.949 (23–26 °C) to −0.278 (23–47 °C) in the LBT and from 0.970 (23–26 °C) to −1.31 (41–50 °C) in the RBT. The *p*-value for BF ranged from < 0.001 (20–23 °C) to 0.854 (26–44 °C) in the LBT and from <0.001 (20–23 °C) to 0.738 (20–50 °C) in the RBT. The r values for ST at set temperature intervals on the carbon mat ranged from 0.967 (20–23 °C) to 0.078 (20–50 °C) in the LBT and from 0.988 (20–23 °C) to 0.064 (20–50 °C) in the RBT. The *p*-value for ST ranged from <0.001 (20–23 °C) to 0.780 (20–50 °C) in the LBT and from <0.001 (20–23 °C) to 0.738 (20–50 °C) in the RBT. Both blood flow and skin temperature show significant increases according to the set temperature in the FIR mat. In the initial temperature range (23–26 °C), the r value is very high, suggesting that the set temperature and blood flow/skin temperature are closely related. However, some correlation coefficients decrease as the temperature range increases, indicating that the differences in individual responses increase. The *p*-values vary from ≤0.001 to 0.854, and the statistical significance decreases, especially in the upper range. This suggests that the linear FIR emitted from the carbon mat affects the blood flow and skin temperature measurements.

### 3.3. Changes in Blood Flow and Epidermal Temperature Measured in LBT and RBT When Using the Loess Bio-Ball Mat

Table 5 presents the measurements of blood flow and epidermal temperature in the LBT and RBT using a loess bio-ball mat. As the mat’s set temperature increased from 20 °C to 50 °C, blood flow in the LBT significantly increased by 32.86 mL/min/100 g, from 4.23 ± 1.64 mL/min/100 g to 37.09 ± 6.04 mL/min/100 g. The epidermal temperature in the LBT also increased gradually by 7.62 °C (26.31%), from 28.96 ± 1.52 °C to 36.58 ± 0.42 °C. Similarly, blood flow in the RBT increased by 32.63 mL/min/100 g, from 4.20 ± 1.61 mL/min/100 g to 36.83 ± 6.48 mL/min/100 g. The epidermal temperature in the RBT increased by 7.86 °C (27.31%), from 28.78 ± 1.35 °C to 36.64 ± 0.21 °C. These findings suggest that FIR emitted from the loess bio-ball mat significantly enhances both peripheral blood circulation and skin temperature in individuals with cold feet. The physiological effects of FIR emitted from the loess bio-ball mat appear to be most pronounced under these conditions.

Spearman’s correlation coefficients (r) for the relationship between BF and set temperature ranged from 0.948 (20–23 °C) to 0.860 (20–50 °C) in the LBT and from 0.890 (20–23 °C) to 0.048 (20–50 °C) in the RBT. The *p*-values for BF ranged from <0.001 (20–23 °C) to 0.651 (20–50 °C) in the LBT and from <0.001 (20–23 °C) to 0.800 (20–50 °C) in the RBT. Strong positive correlations were observed at lower temperature intervals (r = 0.948 at 20–23 °C in the LBT), and these weakened at higher temperature intervals (r = 0.860 at 20–50 °C). Similarly, strong correlations (r = 0.890) were observed at 20–23 °C in the RBT, but correlation nearly disappeared at higher temperatures (r = 0.048 at 20–23 °C). For skin temperature, r values ranged from 0.954 (20–23 °C) to 0.047 (20–50 °C) in the LBT and from 0.948 (20–23 °C) to 0.086 (20–50 °C) in the RBT. Very strong correlations were observed at lower temperatures (e.g., r = 0.954 in LBT at 20–23 °C), but these correlations dropped sharply at higher temperatures (e.g., r = 0.047 in LBT at 20–50 °C). This may be due to the enhanced FIR emission from the loess bio-balls which likely facilitate blood flow and skin temperature at lower set temperatures. The *p*-values for ST ranged from <0.001 (20–23 °C) to 0.047 (20–50 °C) in the LBT and from <0.001 (20–23 °C) to 0.651 (20–50 °C) in the RBT. These results indicate that ST changes were statistically significant (*p* < 0.001) at lower temperatures but not at higher temperatures in the RBT (*p* = 0.651). As the mat temperature increased, the strength of correlation weakened, possibly due to the influence of external factors, which may interference with FIR-medicated effects.

### 3.4. Blood Flow and Epidermal Temperature According to Set Temperature in LBT When Using Electric Mat, Carbon Graphene Mat, and Loess Bio-Ball Mat

Figure 4 illustrates the changes in blood flow and epidermal temperature in the LBT in response to different set temperatures of the electric mat, carbon mat, and loess bio-ball mat. For clarity, only the average values of the blood flow and epidermal temperature are presented, excluding standard deviations. As shown in Figure 4, the blood flow at the corresponding temperature exhibited minimal change with the electric mat (green circles), a moderate increase (15.05 mL/min) with the carbon mat (blue circles), and a substantial increase (32.86 mL/min/100 g) with the loess bio-ball mat (red circles). This difference is attributed to the FIR characteristics of each material. When the carbon mat was used, FIR was emitted only from the carbon-coated heating wire, resulting in a relatively low energy output and thus a smaller increase in blood flow. In contrast, the loess bio-balls emitted a significantly higher amount of FIR, particularly in the 9.5–9.8 μm wavelength range, which is known to promote peripheral blood circulation effectively. The loess bio-ball mat induced the largest increase in blood flow (32.86 mL/min/100 g), which was markedly higher than the increase observed with the other two mats. Regarding epidermal temperature, the green, blue, and red lines in Figure 4 represent the temperature changes observed with the electric, carbon, and loess bio-ball mats, respectively. A notable increase in skin temperature was also observed with the loess bio-ball mat, with a recorded rise of 7.62 °C (26.31%).

This significant increase in blood flow and epidermal temperature was attributed to the enhanced FIR emission from the loess bio-balls. These findings suggest that the loess bio-ball mat, through its enhanced FIR emission, is more effective at improving peripheral blood circulation and skin temperature compared to conventional electric or carbon mats. The 3D plot in Figure 4 clearly highlights the superior physiological effects of the loess bio-ball mat across the full range of temperature settings.

### 3.5. Blood Flow and Epidermal Temperature According to the Set Temperature in RBT When Using Electric Mat, Carbon Mat, and Loess Bio-Ball Mat

Figure 5 illustrates the change in blood flow and epidermal temperature in the RBT according to the set temperature when using an electric mat, carbon mat, or loess bio-ball mat. For clarity, only the average values of the blood flow and epidermal temperature are shown, excluding standard deviations. As shown in Figure 5, blood flow at the corresponding set temperature exhibited minimal change with the electric mat (green circles), a moderate increase (14.55 mL/min/100 g) with the carbon mat (blue circles), and a significant increase (32.63 mL/min/100 g) with the loess bio-ball mat (red circles). This difference is attributed to the distinct FIR characteristics of each mat. In the case of a carbon mat, FIR is emitted only from the carbon-coated heating wire, and the overall energy output is relatively low, resulting in a limited increase in blood flow. In contrast, the loess bio-ball mat emits a higher amount of FIR, particularly in the 9.5–9.8 μm wavelength range, which is known to effectively promote blood circulation. The loess bio-ball mat showed the largest blood flow increase of 32.63 mL/min/100 g, which was substantially higher than that observed with the other mats. In terms of epidermal temperature, the green, blue, and lines in Figure 5 represent the changes corresponding to the electric, carbon, and loess bio-ball mats, respectively. A substantial increase in skin temperature was also observed with the loess bio-ball mat, with a recorded rise of 7.86 °C (27.31%).

Among the three mat types, the loess bio-ball mat resulted in the most pronounced physiological response, producing a 32.63 mL/min/100 g increase in blood flow and a 7.86 °C (27.31%) increase in epidermal temperature in the RBT. These effects are likely due to the high FIR emission in the 9.5–9.8 μm wavelength range, which enhances peripheral vasodilation and thermal conduction. In contrast, the electric mat produced negligible change, and the carbon mat yielded only moderate improvements. This significant increase in both blood flow and epidermal temperature with the loess bio-ball mat demonstrates its superior effectiveness, likely due to its enhanced FIR radiation properties.

### 3.6. Scatter Plot Showing the Slope and Offset of Blood Flow in LBT and RBT for 90 Experimental Subjects

Figure 6 presents a scatter plot (slope vs. offset) illustrating the relationship between blood flow and set temperature in the LBT and RBT of 90 participants. In this plot, the slope represents the rate of change in blood flow as the set temperature increases, while the offset represents the estimated blood flow value when the set temperature is zero (y-intercept). In control Group A (LBT-EM and RBT-EM), where the electric mat was used, the circles and triangles are clustered near the origin (slope ≈ 0), with offsets mainly between 0 and 1. This indicates minimal change in blood flow with increasing temperature when using the electric mat. In experimental Group B (LBT-CM and RBT-CM), where the carbon mat was used, the blue circles and triangles are more spread out along the slope axis (0.2 to 1.1) and y-axis (offset: –20~0.62), suggesting a moderate temperature-induced increase in blood flow. In contrast, in experimental Group C (LBT-BM and RBT-BM), where the loess bio-ball mat was used, the black crosses and red triangles are more widely distributed. The slopes range from 0.1 to 1.5, and the offsets vary from approximately –6 to +43. These results suggest that participants with lower initial blood flow (lower offset) experienced greater temperature-related increases in blood flow (higher slope), while those with higher initial flow had smaller rates of change.

The black crosses represent the LBT data (*n* = 30), and the red triangles represent the RBT data. The trend line for LBT-BM (black dashed line) is y=−33.769x+37.858, $R^{2}=0.7390.$ The trend line for RBT-BM (red solid line) is y=−31.268x+36.044, $R^{2}=0.7721.$ These linear regressions suggest an inverse relationship between the initial blood (offset) and the slope of increase, further highlighting the enhanced vascular response induced by FIR emitted from the loess bio-ball mat.

## 4. Discussion

The scatter plot analysis (Figure 6) reveals clear differences in vascular response patterns across the three heating mat types. In the electric mat group (EM), both the slope and offset values are clustered near zero, indicating a negligible increase in blood flow with rising temperature. This finding supports the interpretation that electric heating alone provides minimal stimulation to peripheral circulation. In the carbon mat group (CM), moderate slopes and a wider range of offset values are observed, suggesting a partial FIR-induced vasodilation. However, the most pronounced effect is observed in the loess bio-ball mat group (BM), which shows a broader distribution of slopes (0.1–1.5) and offsets (−6 to ± 43), indicating strong variability in individual responses. Notably, the negative correlation between slope and offset in the BM group (R^2^ = 0.739 for LBT, R^2^ = 0.7721 for RBT) suggests that individuals with initially lower blood flow experienced a more pronounced increase in response to FIR exposure. This pattern supports the hypothesis that FIR emitted from loess bio-balls more effectively stimulates peripheral vasodilation in individuals with compromised baseline circulation (4.23 to 37.09 mL/min/100 g in the LBT; 4.20 to 36.82 mL/min/100 g in the RBT). Therefore, FIR-based interventions may be particularly beneficial for patients with cold extremities or poor peripheral perfusion.

The present study demonstrated that FIR therapy using a loess bio-ball mat significantly increased peripheral blood flow and epidermal temperature in the big toes, suggesting its potential role in enhancing microcirculation. These findings support the hypothesis that FIR emitted from loess bio-balls, when applied to proximal body regions, may improve peripheral blood flow and skin temperature in individuals with cold extremities. This effect is likely mediated by capillary dilation via the thermoregulatory response in the hypothalamus and nitric oxide (NO)-dependent pathways. FIR wavelengths in the range of 8–15 μm (particularly 9.4 μm) have been reported to activate the NO signaling pathway [6]. Specifically, FIR in this range is absorbed by skin and water molecules, inducing molecular vibration and water activation. These biophysical effects increase local tissue temperature and stimulate the eNOS (endothelial nitric oxide synthase) pathway, thereby promoting NO production [32,33]. Elevated NO levels lead to vasodilation and increased peripheral blood flow. Furthermore, FIR wavelengths around 9.4 μm have been shown to stimulate intracellular heat shock proteins (HSPs), which in turn enhance eNOS expression [8]. This mechanism improves vascular endothelial function and promotes vascular smooth muscle relaxation, thereby increasing blood flow in both capillaries and peripheral vessels.

Similar to the findings of Peng et al. [34], the present study observed a significant increase in skin temperature following FIR exposure. However, unlike their study (LBT: 30.8 to 34.8 °C; RBT: 29.6 to 35.3 °C),which measured toe temperature using an FIR lamp and an infrared camera during localized FIR irradiation, the current study recorded significant increases in both blood flow (LBT: 4.23 to 37.09 mL/min/100 g; RBT: 4.20 to 36.83 mL/min/100 g) and skin temperature (LBT: 28.96 to 36.58 °C; RBT: 28.78 to 36.64 °C) in the big toes of individuals with cold feet. This outcome is attributed to the whole-body FIR irradiation while gradually increasing the set temperature of the loess bio-ball mat from 20 °C to 50 °C in 1 °C increments, thereby exposing participants to a larger amount of FIR. Furthermore, it is proposed that the synergistic effect that resulted from the combined effects of elevated core temperature induced FIR at a peak wavelength of 9.4 μm and the activation of the thermoregulatory mechanism. FIR may stimulate endothelial cells in the blood vessels to generate heat and release NO, which relaxes vascular smooth muscle and promotes increase blood flow.

The effects of FIR treatment on foot circulation in diabetic hemodialysis patients have been shown to include significant increases in temperature, pulse, and blood flow in the dorsal artery of the foot [35]. Diabetic patients undergoing hemodialysis (*n* = 58) were randomly assigned to receive FIR therapy (40 min/session, 3 times a week, for 6 months) applied to both dorsalis pedis arteries. The control group (*n* = 27) received conventional dialysis treatment. FIR therapy showed significant positive effects on the temperature, pulse, and blood flow in the dorsalis pedis arteries. Additionally, sensitivity to pain, touch, and pressure significantly improved in the experimental group. In comparison, FIR emitted from loess bio-balls is expected to offer greater convenience and long-term therapeutic effects, as diabetic patients can apply it to the entire body at home during daily life and sleep.

The relationship between IR radiation and body temperature or skin blood flow (reciprocal circulation) has been investigated [36]. After thermal stimulation of the hind foot using infrared light and a warm bath, increases in skin temperature and blood flow were observed in the contralateral paw. Blood flow and temperature rose on both sides, and FIR was found to induce immediate changes. According to the measurement process and findings of this study, FIR transfers energy in the form of electromagnetic radiation, which rapidly promotes blood circulation. Skin temperature increases accordingly, mainly due to enhanced heat conduction. It is expected that a more pronounced therapeutic effect will occur when FIR emitted from a loess bio-ball mat is applied to the entire body.

FIR delivered vis loess bio-ball mats produced the largest increases in toe blood flow and epidermal temperature among the three heating modalities tested. These improvements—likely driven by hypothalamic thermoregulatory activation following FIR-induced molecular vibration—indicate a clinically meaningful enhancement of peripheral microcirculation. Because the mat is non-invasive and can be used conveniently during sleep, it may serve as adjuvant option for patients with peripheral vascular insufficiency. However, our conclusions are tempered by several limitations, including modest sample size, season restriction to winter, and the absence of long-term follow-up. Larger, multicenter trials involving repeated applications and extended observation are needed to evaluate efficacy and safety in the broader population, including individuals with peripheral arterial disease, diabetes, dysmenorrhea, and hypertension, to assess the long-term sustainability of the benefit.

## 5. Conclusions

When the set temperature was increased from 20 °C to 50 °C using an electric mat, changes in blood flow and epidermal temperature in the LBT and RBT were minimal. In contrast, the carbon mat led to moderate increases in both parameters. Notably, the loess bio-ball mat produced the most substantial enhancements. Specifically, blood flow in the LBT increased by 32.86 mL/min/100 g, and epidermal temperature rose by 7.62 °C (26.31%), from 28.96 ± 1.52 °C to 36.58 ± 0.42 °C. Similar trends were observed in the RBT, with blood flow increasing by 32.64 mL/min/100 g and epidermal temperature rising by 7.86 °C (27.31%).

Far-infrared radiation (FIR) emitted from loess bio-balls, in the wavelength range of 9.4–9.8 μm, is hypothesized to activate water molecules and slightly elevate core body temperature. This initiates thermoregulatory response that enhances peripheral blood flow. Improved circulation, in turn, raises the skin temperature in peripheral regions, particularly in areas rich in capillaries such as the toes. Therefore, FIR therapy using loess bio-balls may serve as a complementary therapeutic modality for managing poor circulation.

## Figures and Tables

**Figure 1 biomedicines-13-01759-f001:**
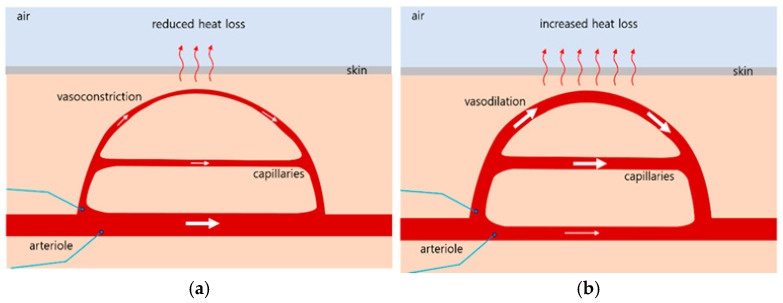
Hypothalamic thermoregulatory mechanism. (**a**) Vasoconstriction response when the body’s core temperature falls below 37 °C, leading to reduced capillary blood flow and minimized heat loss. (**b**) Vasodilation response is triggered when the body’s core temperature increases beyond normal levels (e.g., above 37 °C), resulting to increased capillary blood flow and enhanced heat dissipation.

**Figure 2 biomedicines-13-01759-f002:**
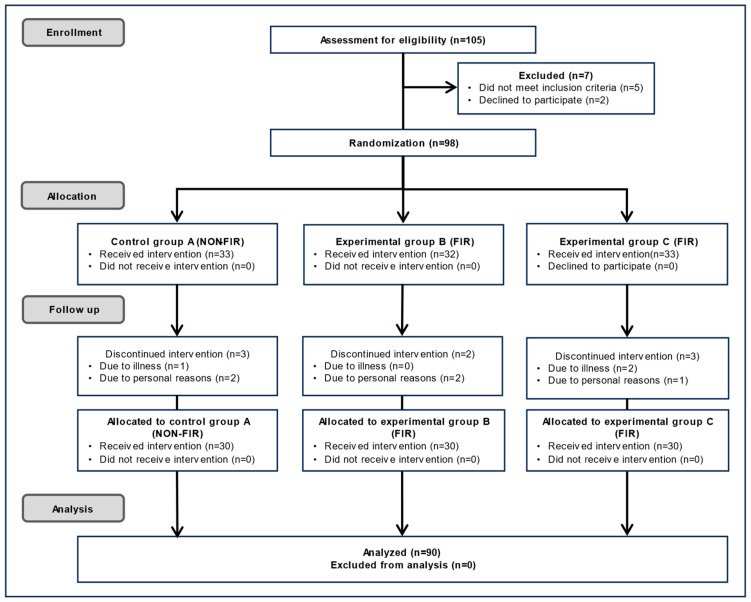
CONSORT diagram of enrollment, participation, and experimental data analysis. (Adapted with permission from the CONSORT 2010 statement [31]).

**Figure 3 biomedicines-13-01759-f003:**
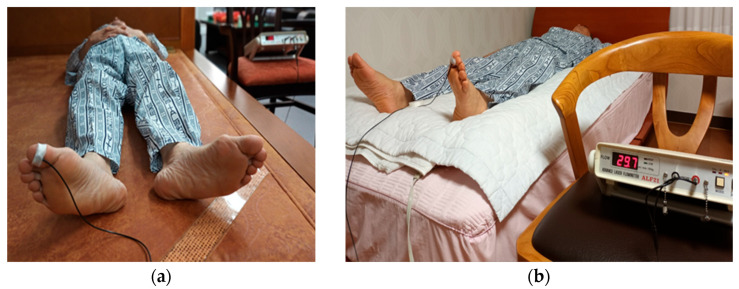
Photographs showing the placement of electrodes used for for measuring blood flow. Reflective electrodes were attached to right and left big toes (RBT, LBT). (**a**) The participant lay on a loess bio-ball bed with electrode attached to the big toe of the right foot. (**b**) A cushion was placed on a loess bio-ball bed, and the participant lay down with an electrode attached to the left big toe.

**Figure 4 biomedicines-13-01759-f004:**
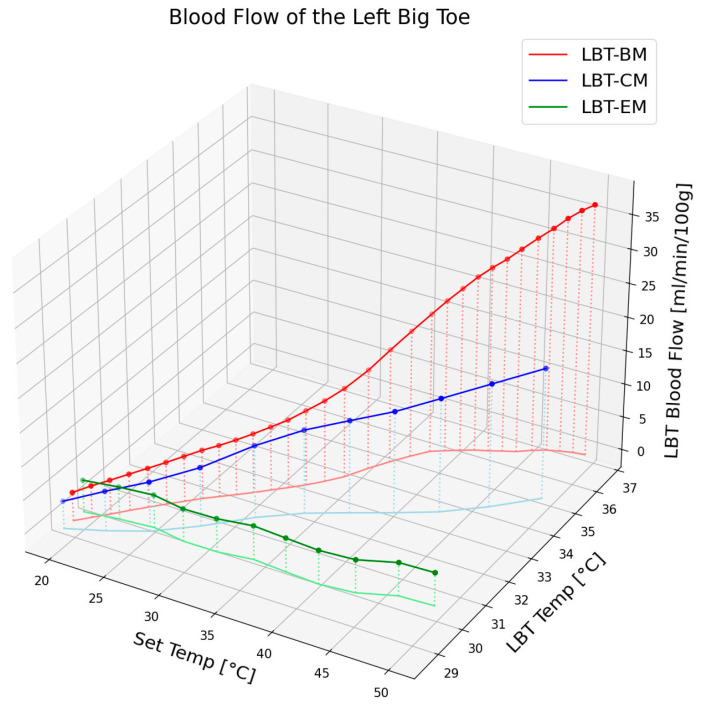
Changes in blood flow and epidermal temperature in the LBT according to set temperature, measured in 90 participants using an electric mat (EM), a carbon mat (CM), and a loess bio-ball mat (BM). The loess bio-ball mat shows the highest increase in both parameters, indicating superior FIR-induced physiological response compared to the other mats.

**Figure 5 biomedicines-13-01759-f005:**
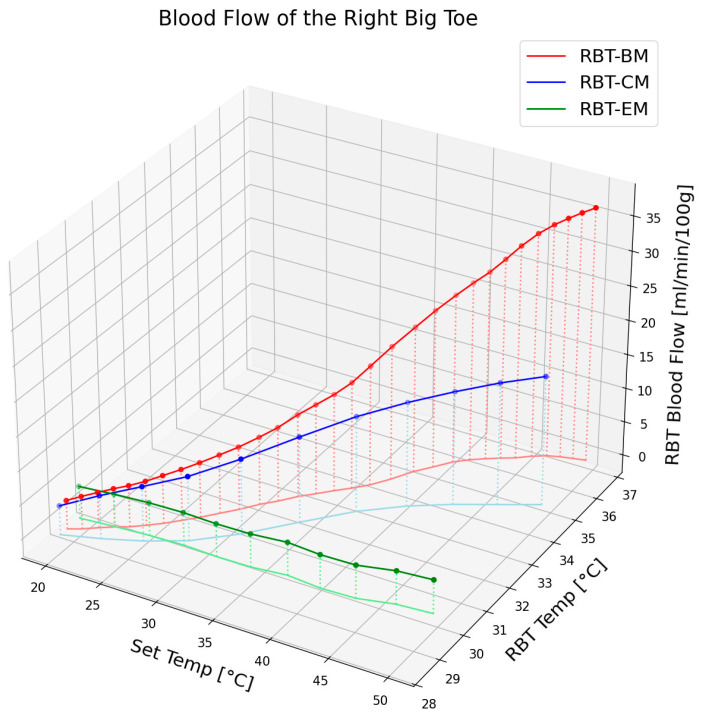
Changes in blood flow and epidermal temperature in RBT according to set temperature when using an electric mat (EM), a carbon mat (CM), and a loess bio-ball mat (BM). The data represent average values from 90 participants. The loess bio-ball mat shows the greatest increase in both parameters compared to the electric and carbon mats.

**Figure 6 biomedicines-13-01759-f006:**
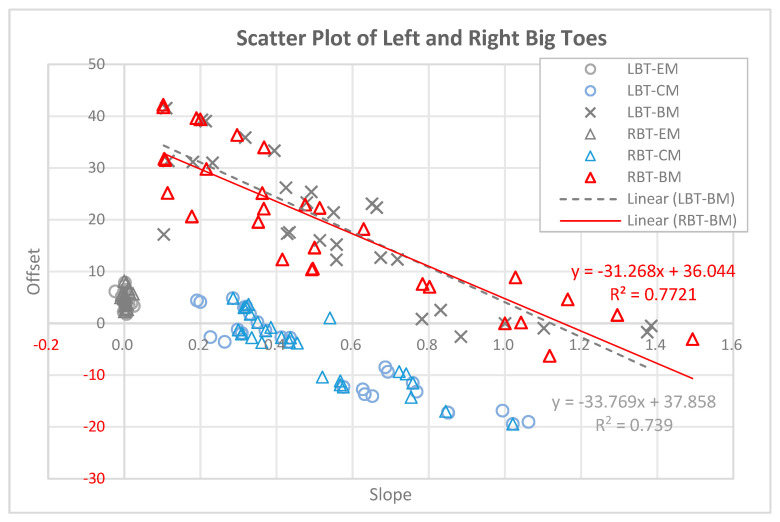
Scatter plot showing the slope and offset of blood flow in LBT and RBT of 90 participants, based on three types of heating mats: electric (EM), carbon (CM), and loess bio-ball mat (BM). Slope indicates the rate of increase in blood flow with rising temperature, while offset represents the estimated blood flow at the staring temperature. Linear trend lines for LBT-BM and RBT-BM show an inverse relationship between slope and offset, indicating greater responsiveness in individuals with initially low blood flow.

**Table 1 biomedicines-13-01759-t001:** Chemical composition (%) of loess powder obtained by XRD measurement.

Chemical Substance	Composition Ratio (%)
SiO_2_	72.3262
Al_2_O_3_	15.9731
Fe_2_O_3_	5.9951
K_2_O	2.1878
MgO	1.0557
TiO_2_	0.9764
CaO	0.4570
P_2_O_5_	0.3812
MnO	0.1914
Na_2_O	0.1912

**Table 2 biomedicines-13-01759-t002:** Participant demographics and baseline characteristics.

Variable	Group A (*n* = 30) (Control)	Group B (*n* = 30) (Line-Emitting FIR)	Group C (*n* = 30) (Area-Emitting FIR)	Effect Size (Cohen’s d)
Age (years)	57.44 ± 8.53	59.24 ± 6.82	60.17 ± 10.35	A vs. B: 0.23 A vs. C: 0.30
Height (cm)	165.12 ± 8.65	164.90 ± 9.38	163.70 ± 9.19	A vs. B: 0.02 A vs. C: 0.16
Weight (kg)	63.70 ± 9.19	65.41 ± 9.03	63.66 ± 6.79	A vs. B: 0.19 A vs. C: 0.00
BMI (kg/m^2^)	23.36 ± 3.26	24.05 ± 2.40	23.76 ± 2.54	A vs. B: 0.23 A vs. C: 0.12
Gender (M/F)	14/16 (46.7%/53.3%)	15/15 (50.0%/50.0%)	16/14 (53.3%/46.7%)	
Skin Temp - LBT (°C)	29.36 ± 1.61	28.61 ± 1.50	28.96 ± 1.52	A vs. B: 0.48 A vs. C: 0.25
Skin Temp - RBT (°C)	29.26± 1.69	28.52 ± 1.24	28.78 ± 1.35	A vs. B: 0.48 A vs. C: 0.28
Blood Flow—LBT (ml/min/100 g)	4.77 ± 1.44	4.12 ± 2.22	4.23 ± 1.64	A vs. B: 0.34 A vs. C: 0.32
Blood Flow—RBT (ml/min/100 g)	4.72 ± 1.47	4.26 ± 2.29	4.20 ± 1.61	A vs. B: 0.23 A vs. C: 0.33

Note: Cohen’s d values are approximated for illustration.

**Table 3 biomedicines-13-01759-t003:** Blood flow and epidermal temperature in the big toes at different set temperatures when using an electric mat.

Temp. [°C]		20	23	26	29	32	35	38	41	44	47	50
L B T	Blood Flow [ml/min/100 g]	4.77 ± 1.44	4.84 ± 1.46	4.89 ± 1.47	4.87 ± 1.44	4.96 ± 1.49	5.02 ± 1.46	5.03 ± 1.46	5.00 ± 1.43	4.88 ± 1.44	4.92 ± 1.41	4.87 ± 1.46
	Skin Temp. [°C]	29.36 ± 1.61	29.48 ± 1.66	29.57 ± 1.66	29.42 ± 1.81	29.43 ± 1.78	29.56 ± 1.69	29.49 ± 1.73	29.43 ± 1.76	29.53 ± 1.82	29.86 ± 1.65	29.90 ± 1.60
R B T	Blood Flow [ml/min]	4.72 ± 1.47	4.80 ± 1.49	4.83 ± 1.53	4.81 ± 1.55	4.82 ± 1.53	4.85 ± 1.54	4.82 ± 1.50	4.87 ± 1.50	4.82 ± 1.51	4.84 ± 1.53	4.88 ± 1.55
	Skin Temp. [°C]	29.26 ± 1.69	29.35 ± 1.71	29.42 ± 1.70	29.46 ± 1.73	29.43 ± 1.74	29.44 ± 1.68	29.55 ± 1.68	29.47 ± 1.62	29.51 ± 1.63	29.73 ± 1.64	29.80 ± 1.59

**Table 4 biomedicines-13-01759-t004:** Blood flow and epidermal temperature in the big toes at the set temperatures when using a carbon graphene mat.

Temp. [°C]		20	23	26	29	32	35	38	41	44	47	50
L B T	Blood Flow [ml/min/100 g]	4.12 ± 2.22	5.88 ± 2.63	7.31 ± 3.23	8.68 ± 3.27	10.66 ± 4.27	12.37 ± 5.58	13.69 ± 5.81	15.01 ± 5.71	16.77 ± 5.15	18.14 ± 5.32	19.17 ± 5.44
	Skin Temp. [°C]	28.61 ± 1.50	28.97 ± 1.36	29.41 ± 1.29	30.10 ± 1.41	30.94 ± 1.66	31.59 ± 1.88	32.07 ± 2.02	32.53 ± 1.98	33.03 ± 1.83	33.72 ± 1.51	34.56 ± 1.26
R B T	Blood Flow [ml/min/100 g]	4.26 ± 2.29	6.34 ± 2.83	7.99 ± 3.48	9.42 ± 3.69	10.97 ± 4.09	12.52 ± 4.64	13.97 ± 4.66	15.27 ± 4.47	16.58 ± 4.17	17.88 ± 4.07	18.81 ± 4.25
	Skin Temp. [°C]	28.52 ± 1.24	28.81 ± 1.36	29.18 ± 1.41	29.66 ± 1.34	30.43 ± 1.28	31.42 ± 1.58	32.38 ± 1.93	33.09 ± 2.00	33.64 ± 1.87	34.10 ± 1.63	34.56 ± 1.34

**Table 5 biomedicines-13-01759-t005:** Blood flow and epidermal temperature in the big toes at the set temperatures when using a loess bio-ball mat.

Temp. [°C]		20	23	26	29	32	35	38	41	44	47	50
L B T	Blood Flow [ml/min/100 g]	4.23 ± 1.64	5.45 ± 2.13	6.77 ± 2.53	8.24 ± 3.10	10.20 ± 3.82	13.32 ± 4.96	18.89 ± 8.48	24.34 ± 9.80	28.77 ± 9.72	33.07 ± 8.30	37.09 ± 6.04
	Skin Temp. [°C]	28.96 ± 1.52	29.88 ± 1.20	30.75 ± 1.00	31.51 ± 1.01	32.29 ± 1.09	33.25 ± 1.21	34.67 ± 1.42	35.41 ± 1.40	35.83 ± 1.18	36.34 ± 0.58	36.58 ± 0.42
R B T	Blood Flow [ml/min/100 g]	4.20 ± 1.61	5.65 ± 2.34	7.04 ± 2.93	8.60 ± 3.73	10.88 ± 4.31	14.23 ± 5.68	19.57 ± 9.61	24.63 ± 10.61	29.27 ± 9.33	33.86 ± 7.64	36.83 ± 6.48
	Skin Temp. [°C]	28.78 ± 1.35	29.34 ± 1.25	29.97 ± 1.29	30.90 ± 1.49	31.92 ± 1.77	32.86 ± 1.93	33.87 ± 1.68	35.14 ± 1.29	35.79 ± 1.06	36.38 ± 0.32	36.64 ± 0.21

## Data Availability

Data contain sensitive personal and physical information about the study participants and are available from the corresponding author upon reasonable request.

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
