# Peer review of "Improvement of Blood Flow and Epidermal Temperature in Cold Feet Using Far-Infrared Rays Emitted from Loess Balls Manufactured by Low-Temperature Wet Drying Method: A Randomized Trial"

_biomedicines, 2025, doi:10.3390/biomedicines13071759_

Round 1

Reviewer 1 Report

Comments and Suggestions for Authors

The study of Yong Il Shin, MinSeok Kim, Yeong Ae Yang, Gye Rock Jeon, Jae Ho Kim, Yeon Jin Choi, Woo Cheol Choi, Jae Hyung Kim "Improvement of Blood Flow and Epidermal Temperature in Cold Feet Using Far-Infrared Rays Emitted from Functional Loess Bio-Balls" is devoted to an important problem of treating microcirculatory disorders of skin vessels. This is a wide group of pathologies, including diabetic foot, Raynaud's disease and other conditions that significantly worsen the quality of life or lead to disability. There is no effective therapy for microcirculatory disorders today, which increases the relevance of the work. The idea proposed by the authors is quite new, and the proposed hypothesis about the participation of selected resonant frequencies of water in the effects of IR on skin microcirculation looks interesting. Among the advantages of the work, I would like to note the following. Medical trials were performed at a high level. A detailed study design is provided. The study itself was performed as a "blind test". Detailed demographic data on patients are provided. Sham control was performed close to the experimental groups. The sample sizes are justified by the statistical power of the criteria used. A huge array of experimental data has been analyzed, and they are presented in the form of very concise and understandable tables and graphs. The quantitative data indicated in the conclusions are fully justified by the results. I believe that the work can be published after revision.

Main comments

The introduction says little about loess powders and their place in the present study. I believe that it is desirable to briefly give the prerequisites for their selection for the study.

The introduction lacks a summary sentence about the purpose of the study. It is advisable for the authors to outline the working hypothesis explaining the effects being studied at the end of the introduction so that the reader can better understand the general idea of the study.

Materials and methods:

It would be desirable to add a description of the source loess powders: initial source? chemical composition? how were they modified during the preparation of the balls? (equipment used, heating time, pressing, etc.).

In addition, it is desirable to add representative IR spectra of the original powders and the obtained balls.

An emission spectrum of loess bio-ball is also desirable to provide.

Perhaps bio-ball terminology It is worth clarifying, focusing on the composition of the beads. The target of IR is not necessarily water in living cells and/or living thinks did not used in balls production. Alternatively, the authors should clearly justify the choice of terminology.

In section 2.3 only the systemic mechanism of vascular lumen regulation is described. However, there are local mechanisms of regulation, including TRPV receptor-mediated axon reflex, NO-dependent vasodilation (10.1113/jphysiol.2010.195511, 10.1152/japplphysiol.00690.2009, 10.3390/biology13090685 etc.). At the same time, in the case of regulation of skin capillary lumens, local mechanisms can be expressed significantly. It is advisable for the authors to supplement this section, and also to pay attention to this when discussing obtained results.

Section 2.4 must clearly indicate that all manipulations were in accordance with the Declaration of Helsinki.

For LDF measurements, it is advisable to specify the registration details: number of light guides, measurement mode, time of one measurement, post-processing of the signal (averaging of several measurement results, “zero subtraction”, etc., if they took place).

In Section 2.7, it is necessary to specify methods for testing the normality of the distribution. If the distributions of sample values differ from normal, then the Spearman correlation criterion will be more accurate.

Results of Tables 2 and 3 show increasing of skin temperature >32-34 ° C, which in itself is a powerful vasodilating effect. Local heating is a classic functional test in the study of microcirculation in health and pathology (10.1134/S0006350923060180 etc.). The growth of BF can be caused by more efficient heating of the skin. To test the hypothesis of the participation of the selected IR frequencies in the regulation of the skin blood flow velocity, it is advisable to perform control measurements with heating according to the same time profile as in Table 3, but without bio-ball ma t. The possible contribution of the axon reflex should be discussed.

LDF signal is proportional not only to the flow rate, but also to the number of blood cells per unit of blood volume, so I consider the more generally accepted BPU (Blood Perfusion Units) to be a more correct form of presentation.

The discussion should discuss the contribution of local mechanisms of regulation of the skin capillary blood flow. In case of the assumption of a central mechanism, it is necessary to assume and justify at least partially the chain of signal transmission along the path "skin sensors - hypothalamus". However, the contribution of the systemic mechanism can be assumed, having data on the change in temperature/microcirculation speed in a skin area remote from the experimental effect. It is advisable for the authors to discuss infrared light-induced endothelium-dependent vasodilation as a potential mechanism of the observed effects (10.3389/fphys.2023.1219998 etc.).

When discussing the molecular mechanisms of IR on microcirculation, it is advisable for the authors to propose a hypothetical mechanism linking the well-known pathways of vasodilation/vasoconstriction regulation with water absorption in the resonant IR region (for example, ROS generation, conformational changes in macromolecules and their hydration shells, or other).

As a research prospect, I propose to discuss the possibility of assessing the amplitude-frequency characteristics of microcirculation perfusion rate oscillations in different areas of the skin to obtain additional information about the targets and mechanisms of regulation of skin blood perfusion rate during exposure by IR with different wavelengths.

Minor corrections

line 60: in vivo should be italic

line 61: Resonance frequency values of water molecules in IR from literature data should be indicated.

line 71: Authors are encouraged to add an approximate range of values defined as low energy IR to make the manuscript more understandable to researchers from other fields.

line 149: m 2 should be superscript

line 322: missed space

Correlation plots (data from Tables 1-3) should be added to supplementary.

Best regards

Author Response

Main comments

Comment 1: 82-91,

 The introduction says little about loess powders and their place in the present study. I believe that it is desirable to briefly give the prerequisites for their selection for the study.

Response1: 82-91, 111-133

In this study, the characteristics of loess powder and selected loess bio-balls manufactured by a low-temperature wet drying method were described.    

Comment 2: 

The introduction lacks a summary sentence about the purpose of the study. It is advisable for the authors to outline the working hypothesis explaining the effects being studied at the end of the introduction so that the reader can better understand the general idea of the study.

Response 2: 82-91

In the introduction, the prerequisites for selecting loess powder for the study were described.

Materials and methods:

Comment 3:

It would be desirable to add a description of the source loess powders: initial source? chemical composition? how were they modified during the preparation of the balls? (equipment used, heating time, pressing, etc.). In addition, it is desirable to add representative IR spectra of the original powders and the obtained balls. An emission spectrum of loess bio-ball is also desirable to provide.

Response 3: 111-132

 1) The initial source of the loess powder, chemical composition, how it was transformed during the manufacture of the loess powder, equipment used, heating time, and pressurization were described.

2) The absorption spectrum obtained from the original powder and the powder at high temperatures has been described in detail in the already published paper, so instead of describing the content and presenting the figure, it has been replaced with references [15].

3) The emission spectrum of the loess bioball has also been replaced with references, as it has already been published in the material research [15].

Comment 4: Perhaps bio-ball terminology It is worth clarifying, focusing on the composition of the beads. The target of IR is not necessarily water in living cells and/or living thinks did not used in balls production. Alternatively, the authors should clearly justify the choice of terminology.

Response 4: Most loess has low viscosity, so it is baked at over 1000 degrees to make loess balls. Our research on loess material through XRD and IR absorption measurements showed that loess powder heated at high temperatures is denatured and loses all of the useful properties of raw loess. Therefore, it has completely different properties from raw loess or loess manufactured by drying at low temperatures. We are manufacturing loess-bio-balls at low temperatures and conducting research on them in the same way as making pills made from herbal medicine in oriental medicine. All loess balls except ours are baked at over 1000℃and therefore have properties different from raw loess, like bricks. To distinguish them from loess balls manufactured at high temperatures, we call them loess balls made by a low-temperature wet-drying method or loess bio-balls.

Comment 5:

In section 2.3 only the systemic mechanism of vascular lumen regulation is described. However, there are local mechanisms of regulation, including TRPV receptor-mediated axon reflex, NO-dependent vasodilation (10.1113/jphysiol.2010.195511, 10.1152/japplphysiol.00690.2009, 10.3390/biology13090685 etc.). At the same time, in the case of regulation of skin capillary lumens, local mechanisms can be expressed significantly. It is advisable for the authors to supplement this section, and also to pay attention to this when discussing obtained results.

Response 5: 150-243

Systemic and local thermoregulatory mechanisms in the human body is added in section 2.4.

2.4.1 Mechanisms of thermoregulation in the body

2.4.2 TRPV1 receptor-mediated axon reflexes

2.4.3 NO-dependent vasodilation

Conmment 6:

Section 2.4 must clearly indicate that all manipulations were in accordance with the Declaration of Helsinki.

Response 6: 256-257

This study was conducted in accordance with the principles of the Declaration of Helsinki.

Comment 7:

For LDF measurements, it is advisable to specify the registration details: number of light guides, measurement mode, time of one measurement, post-processing of the signal (averaging of several measurement results, “zero subtraction”, etc., if they took place).

Response 7: 325-329, 333-336

The registration specifications of the measuring instrument used for DF measurement and skin temperature measurement are described in detail.

Comment 8:  

In Section 2.7, it is necessary to specify methods for testing the normality of the distribution. If the distributions of sample values differ from normal, then the Spearman correlation criterion will be more accurate.

Response 8: 357-362, 375-384, 400-415, 433-453

In Section 2.8, the method for testing normal distribution and Spearman correlation criterion were described. And these tests and Spearman correlation coefficients were obtained for the measured values.

Comment 9:

Results of Tables 2 and 3 show increasing of skin temperature >32-34 ° C, which in itself is a powerful vasodilating effect. Local heating is a classic functional test in the study of microcirculation in health and pathology (10.1134/S0006350923060180 etc.). The growth of BF can be caused by more efficient heating of the skin. To test the hypothesis of the participation of the selected IR frequencies in the regulation of the skin blood flow velocity, it is advisable to perform control measurements with heating according to the same time profile as in Table 3, but without bio-ball mat. The possible contribution of the axon reflex should be discussed.

Response 9: 559-563

In the discussion, it was described that capillaries are expanded through the hypothalamic thermoregulatory mechanism and vasodilation by NO, thereby improving blood flow and increasing skin temperature in the extremities of patients with cold feet.

Comment 10:

LDF signal is proportional not only to the flow rate, but also to the number of blood cells per unit of blood volume, so I consider the more generally accepted BPU (Blood Perfusion Units) to be a more correct form of presentation.

Response 10:  

Throughout the paper, blood flow was expressed in units of BPU (ml/min/100g).

Comment 11:

The discussion should discuss the contribution of local mechanisms of regulation of the skin capillary blood flow. In case of the assumption of a central mechanism, it is necessary to assume and justify at least partially the chain of signal transmission.

Response 11: 559-573

The contribution of local mechanisms to the regulation of cutaneous capillary blood flow is discussed in detail.

PS:

  1. In addition, the entire paper was reviewed and the content was revised to make it easier for readers to understand, and the English version was also scientifically and meticulously proofread.

  1. I learned a lot in the process of revising the content based on the reviewer's appropriate comments. I would like to express my sincere gratitude for this.

Reviewer 2 Report

Comments and Suggestions for Authors

The manuscript investigates the effects of far infrared (FIR) rays from loess bio-balls on blood flow and epidermal temperature in individuals with cold feet, using a randomized controlled trial. The study premise is promising. Below are some recommendations for improvement from it current form.

  1. For the study design part, provide a detailed description of the randomization process, including sequence generation and allocation concealment.
  2. Authors should specify the instruments used for blood flow and temperature measurements, including their accuracy and calibration as for the measurement of blood flow and epidermal temperature.
  3. You need parameters to support the statistical power calculation by including a complete power calculation with all relevant parameters.
  4. Provide detailed descriptions of figures 4 and 5, including baseline and post intervention values, statistical tests, and p-values.
  5. Update table 1 formatting and include a comprehensive demographic data. Present quantitative comparisons such as mean ± SD, and effect sizes for all groups.
  6. In the mechanistic discussion part, consider strengthening it by linking FIR absorption to specific physiological outcomes observed in the study. Also address potential confounders and discuss limitations such as generalizability to other populations or conditions.

Author Response

The manuscript investigates the effects of far infrared (FIR) rays from loess bio-balls on blood flow and epidermal temperature in individuals with cold feet, using a randomized controlled trial. The study premise is promising. Below are some recommendations for improvement from it current form.

  1. For the study design part, provide a detailed description of the randomization process, including sequence generation and allocation concealment.

Response 1: 293-320

The study administrator was in charge of the random assignment sequence. Participants were randomly assigned to one of the following three groups using a sealed envelope random assignment method containing application forms for the experiment of types A, B, and C. Group A used an electric mat, group B used a carbon mat that emits far-infrared rays in a linear shape from carbon coated with heat wires, and group C used a loess bio mat that emits far-infrared rays in an area shape from loess balls. Cushions and covers were used on top of the three mats so that the participants did not know the type of mat, and the measurers were also unaware of the participants' health history and conditions.

  1. Authors should specify the instruments used for blood flow and temperature measurements, including their accuracy and calibration as for the measurement of blood flow and epidermal temperature.

Response 2: 318-337

Specifications are given for the devices and instruments used to measure blood flow and skin temperature, and the laser Doppler flowmeter is described in detail.

  1. You need parameters to support the statistical power calculation by including a complete power calculation with all relevant parameters.

Response 3: 355-364, 376-385, 401-416, 434-454

The measurement results were analyzed and described using statistical power.

  1. Provide detailed descriptions of figures 4 and 5, including baseline and post intervention values, statistical tests, and p-values.

Response 4: 

The detailed description of Figures 4 and 5 is provided, and the baseline and pre- and post-intervention measurements are described in the raw data in Tables 2, 3, and 4. The statistical analysis of Figures 4 and 5 is described in detail in Tables 2, 3, and 4.

  1. Update table 1 formatting and include a comprehensive demographic data. Present quantitative comparisons such as mean ± SD, and effect sizes for all groups.

Response 5: 272-276

Table 2 provides detailed descriptions and presents demographic data and quantitative representations and Cohen's effect sizes for all groups.

  1. In the mechanistic discussion part, consider strengthening it by linking FIR absorption to specific physiological outcomes observed in the study. Also address potential confounders and discuss limitations such as generalizability to other populations or conditions.

Response 6:  279-313, 622-633

 In the mechanistic considerations section, consider strengthening this section by linking FIR absorption to specific physiological outcomes observed in the study. Also address potential confounding factors and discuss limitations such as generalizability to other populations or conditions.

PS: 1. In addition, the entire paper was reviewed and the content was revised to make it easier for readers to understand, and the English version was also scientifically and meticulously proofread.

  1. I learned a lot in the process of revising the content based on the reviewer's appropriate I would like to express my sincere gratitude for this.

Round 2

Reviewer 1 Report

Comments and Suggestions for Authors

The authors responded to all comments in detail. I believe that the manuscript can be published.

Minor:

Line 234: Please check your superscript letters in new text at proofread stage. For example, Na+ and Ca2+ should be Na+ and Ca2+ .

Best regards

Reviewer 2 Report

Comments and Suggestions for Authors

All concerns from previous round of revision were addressed.